# Effects of Dietary Phosphorus Levels on Growth Performance, Phosphorus Utilization and Intestinal Calcium and Phosphorus Transport-Related Genes Expression of Juvenile Chinese Soft-Shelled Turtle (*Pelodiscus sinensis*)

**DOI:** 10.3390/ani12223101

**Published:** 2022-11-10

**Authors:** Yue Wang, Yiran Geng, Xueying Shi, Siqi Wang, Zhencai Yang, Peiyu Zhang, Haiyan Liu

**Affiliations:** Laboratory of Aquatic Animal Nutrition and Ecology, College of Life Sciences, Hebei Normal University, Shijiazhuang 050024, China

**Keywords:** phosphorus, growth performance, phosphorus utilization, calcium and phosphorus transport, *Pelodiscus sinensis*

## Abstract

**Simple Summary:**

Phosphorus is a vitally important mineral to ensure healthy growth for aquaculture animals, but is also the main contributor to eutrophication. To guarantee green and high-quality development of the aquaculture industry, it is crucial to quantify the dietary phosphorus requirement of cultured species. In this study, we assessed the influence of four experimental diets containing different phosphorus levels on the juvenile Chinese soft-shelled turtle (*Pelodiscus sinensis*). The results indicated that the available phosphorus requirement of juvenile *P. sinensis* was 1.041%. Lower or higher dietary phosphorus level negatively affected growth and feed utilization of the turtles. Furthermore, intake of high-phosphorus diets also led to significantly higher phosphorus discharge into the water body.

**Abstract:**

A 60-day feeding trial was performed to assess the effects of dietary phosphorus levels on growth performance, body composition, phosphorus utilization, plasma physiological parameters and intestinal Ca and P transport-related gene expression of juvenile Chinese soft-shelled turtle (*P*. *sinensis*). Four diets containing available P at graded levels of 0.88%, 1.00%, 1.18% and 1.63% (termed as D0.88, D1.00, D1.18 and D1.63, respectively) were formulated and each diet was fed to turtles (5.39 ± 0.02 g) in sextuplicate. The turtles were randomly distributed to 24 tanks with 8 turtles per tank. The results indicated that final body weight, specific growth rate, feed conversion ratio and protein efficiency ratio performed best in turtles fed 1.00% available P diet. The crude lipids of the whole body exhibited a decreasing trend with the dietary available P, whereas the calcium and phosphorus of the whole body and bone phosphorus showed an opposite tendency. The apparent digestibility coefficient of phosphorus declined with the dietary available P. Turtles fed 1.00% available phosphorus had the highest phosphorus retention ratio compared with other treatments. Simultaneously they had significantly lower phosphorus loss than turtles fed D1.18 and D1.63 and had no differences in this respect from turtles fed a low-phosphorus diet. It was noteworthy that the lowest plasma calcium concentrations, and alkaline phosphatase activities in plasma and liver, were discovered in turtles fed the diet containing 1.63% available phosphorus. In addition, the high-phosphorus diet resulted in significantly down-regulated expression of intestinal phosphorus and calcium transport-related key genes. In conclusion, the available phosphorus requirement of juvenile *P*. *sinensis* was determined at 1.041% (total phosphorus was 1.80%) based on quadratic regression of weight gain rate, and excessive dietary phosphorus stunted turtle growth possibly via inhibiting intestinal calcium absorption.

## 1. Introduction

The Chinese soft-shelled turtle (*P. sinensis*) is one of the significant freshwater aquaculture animals in China owing to its fast growth, high nutritive and economic value [1], and the annual production has held steady at more than 300 thousand tonnes over the past years [2,3,4]. Under the dual pressure of aqueous ecological environment protection and the rapidly increasing price of feedstuffs, there is a desperate need to develop a high-efficiency and environmentally friendly compound feed for this species based on its nutrient requirements. Although some literature exists on its protein, carbohydrate, lipid, vitamins, and mineral requirements [5,6,7,8,9,10], little information is available regarding dietary phosphorus requirement of this animal.

Phosphorus is commonly regarded as one of the foremost minerals needed by aquaculture animals. It is a key component of nucleic acids, biomembranes and coenzymes, and plays a vital role in energy metabolism, oxygen transport in red cells and maintenance of normal acid-base equilibrium [11]. The quantity of phosphorus absorbed from the surrounding water is negligible in aquatic animals; hence, they depend largely on phosphorus of dietary origin [12]. Most studies on dietary phosphorus requirements in aquatic animals have revealed that inadequate dietary phosphorus could cause lessened growth performance and feed utilization, and result in irregular skeletal development [13,14,15,16]. Importantly, a high phosphorus diet may also reduce growth performance, feed utilization, and survival in some aquatic animals [17,18,19], as well as increase phosphorus discharge into water. Therefore, it is of vital importance to ascertain the phosphorus requirement of *P. sinensis* to ensure efficient and healthy aquaculture conditions.

Phosphorus and calcium are two critical minerals for skeletal development in the form of the Ca-P complex, hydroxyapatite [20]. Studies in mammals have indicated that intestinal phosphorus absorption could be enhanced by 1, 25(OH)_2_D_3_ via the binding vitamin D receptor (VDR) [21]. Previous studies have revealed that VDR knockout in the intestine of mice lessens phosphorus absorption efficiency by 50% [22,23]. Study in mammals also has proved that intestinal calcium transport was enhanced by a low phosphorus diet [24]. When the dietary calcium level is normal or low, active transcellular transport is regarded as the predominant calcium absorption mechanism in the intestine [25,26] and this transcellular transport involves three steps. The transient receptor potential vanilloid 6 (TRPV6) is considered a pivotal molecule in the active transcellular entry of calcium through the apical membrane in the gut [27], and the disruption of the TRPV6 in mice led to marked disturbance of calcium homeostasis [28]. The cytoplasmic Ca^2+^ then binds to the protein calbindin and moves to the basolateral membrane [29]. Finally, a Ca pump (plasma membrane Ca^2+^ transporting ATPase) and Na-Ca exchanger (NCX1) on the basolateral membrane extrude the Ca^2+^ to the outside of the cell [30]. In addition, dietary phosphorus level is a main adjustor of intestinal phosphorus absorption in aquatic animals [11]. The phosphorus absorption rate decreased with dietary phosphorus concentration by means of dynamic regulation of the expression of NaPi-IIb transporters located in the intestinal apical membrane for most teleosts [31,32]. Earlier studies found that whole-body calcium content increased with dietary phosphorus concentration in *Myxocyprinus asiaticus* [33], *Pelteobagrus fulvidraco* [34] and *Ctenopharyngodon idella* [35], which suggested that calcium metabolism in the body might be influenced by dietary phosphorus level.

In teleosts, dietary total phosphorus requirement range from 0.29–1.23%, and most fishes converged on 0.5–1.0% [11]. For crustaceans, however, the requirements are more than 1.30%, such as 1.84% in Indian white shrimp (*Fenneropenaeus indicus*) [36], 2.09–2.20% in postlarval *Litopenaeus vannamei* [37], 1.59–1.68% in swimming crab (*Portunus trituberculatus*) [16], 1.16–1.51% (available phosphorus) in Chinese mitten crab (*Eriocheir sinensis*) [38], and 1.39–1.43% in red swamp crayfish (*Procambarus clarkia*) [39]. Chinese soft-shelled turtle has two big shells, namely carapace and plastron, and 65% of the shell is mineral with calcium phosphate as the predominant compound [40]. Therefore, the phosphorus requirement should be higher than finfish and close to crustaceans. In addition, a study on calcium requirement in the red-eared slider turtle (*Pseudemys scripta elegans*) indicated that the best growth performance was achieved in turtles fed 2.0% calcium and 1.2% phosphorus [41]. Likewise, better growth performance was obtained in Chinese soft-shelled turtle fed 5.7% calcium and 3.0% phosphorus [40]. Therefore, the objective of this study was to evaluate the dietary optimal phosphorus requirement and its effect on growth, body composition, physiological status and relative expression of intestinal phosphorus and calcium transport-related key genes in juvenile Chinese soft-shelled turtle (*P. sinensis*).

## 2. Materials and Methods

### 2.1. Animal Ethics

Animal care and all experimental procedures on experimental animal were conducted with permission from the Institutional Animal Care and Use Committee of Hebei Normal University (198012, Shijiazhuang, Hebei, China).

### 2.2. Experimental Diets

In the current study, four isonitrogenous (40.80% crude protein), isoenergetic (19.50 MJ/kg) and iso-calcic (2.50%) semi-practical diets were designed to incorporate exogenous inorganic monocalcium phosphate (MCP) at doses of 0%, 2%, 4% and 8% on a dry matter basis by adjusting the microcrystal cellulose and limestone content. The range of dietary total phosphorus was designed to be from 1.20% to 3.20% according to the description in the Introduction section. The crude protein (40.80%) and carbohydrate contents (16%) covered nutrient requirements of juvenile Chinese soft-shelled turtle according to previous studies [8,42]. The experimental diets utilized Russian white fishmeal, chicken meal, casein and wheat gluten as dominant protein sources, and used pregelatinized corn starch as the carbohydrate source and fish oil as the lipid source. Yttrium oxide was supplemented at 0.1% as an inert marker for the nutrient digestibility analyses [43]. All ingredients were step-by-step mixed, ground sufficiently and sifted (80 mesh); then 4% fish oil was mixed thoroughly with the powder and passed through a 60-mesh sieve three times. Then the mixture was homogenized thoroughly with distilled water (30% of the powder mass) in a blender to form a stiff dough. The dough was then extruded into soft pellets with 2.0 mm diameter using a pelletizer without additional heating in the die (Youyi Machinery Factory). The wet pellets were preserved in airtight plastic bags at −20 °C until use. The diet formulation and proximate chemical composition are illustrated in Table 1. The available phosphorus content in the diets was 0.88%, 1.00%, 1.18% and 1.63%, respectively, computed by the total phosphorus content multiplied by apparent phosphorus digestibility.

### 2.3. Turtles and Feeding Management

Juvenile Yellow River strain Chinese soft-shelled turtles (*P. sinensis*) (mean carapace length: 24.59 mm) were purchased from a farm in Yutian, TangShan, China and transported to the Laboratory of Aquatic Animal Nutrition and Ecology, College of Life Sciences, Hebei Normal University, Shijiazhuang, China. Upon arrival, the turtles were sanitized in 0.05% potassium permanganate for 10 min. As shown in Figure 1, the turtles were then stocked in cylindrical fiberglass tanks (diameter: 60 cm, height: 70 cm) equipped with a perforated metal partition plate. A filter screen was hung as shelter in each tank to reduce possibility of aggressive bites among the turtles. The turtles were then acclimated to the culture environment for 10 days and fed with a commercial feed from Hebei Haitai company (Hebei Haitai Tech. Ltd., Shijiazhuang, China). At the commencement of the feeding trial, 192 healthy turtles (initial body weight: 5.39 ± 0.02 g), after complete deprival of food for 24 h, were randomly placed in 24 tanks with 8 turtles in each tank. And six tanks were randomly allocated to each diet (*n* = 6). During the feeding trial, turtles were manually fed with the four experimental diets to apparent satiation three times a day (8:00, 12:30 and 17:30) for 60 d. After the thirty-minute feeding, feces and residual feed were siphoned out. Subsequently, the feed residue was dried and weighed to correct for the feed intake (FI) and feed conversion ratio (FCR). About 30% of the water by volume was exchanged in each tank and the operations were conducted twice a day. The incandescent light bulb was turned on when residual pellets/feces were removed and tank water was exchanged and the culture room was kept dark using black tarpaulin on windows at other time periods. The water temperature was maintained at 30.0 ± 0.1 °C using a temperature controller, and the pH was at 7.5 ± 0.1.

### 2.4. Apparent Digestibility Measurement

The apparent digestibility coefficients (ADCs) of dry matter and phosphorus were measured applying yttrium oxide (Y_2_O_3_) as an inert marker through the indirect digestibility approach. The feces collection was launched at two weeks after the commencement of the feeding trial and lasted for six weeks. The fresh feces were gently obtained by siphoning with plastic dropper at 3–4 h post each meal, rinsed quickly with distilled water and kept at −20 °C. Afterwards, the fecal samples were freeze-dried for the determination of the ADCs of dry matter and total phosphorus. The fecal materials collected from two tanks each day were pooled as one sample (four fecal samples per treatment). To measure the Y_2_O_3_ content in the diet and feces, the samples were subjected to acid digestion and determined by inductively coupled plasma source mass spectrophotometer (X Series 2 ICP-MS) (Thermo Electron Corporation, Waltham, MA, USA) in College of Life Sciences, Hebei Normal University (Shijiazhuang, Hebei, China) based on a previous study [44]. The dry matter and phosphorus contents were measured according to the following protocol described in the chemical analysis section. The ADCs of dry matter and phosphorus were computed as follows:(1)$$\mathrm{ADC}\;=100\times \left(1-\frac{N_{f}}{N_{d}}\times \frac{Y_{d}}{Y_{f}}\right)$$
where ADC is the apparent digestibility coefficient of the nutrient (%), Y_d_ is the Y_2_O_3_ content in the diet (g kg^−1^); Y_f_ is the Y_2_O_3_ content in the feces (g kg^−1^); N_d_ is the nutrient content in the diet (g kg^−1^); N_f_ is the nutrient content in the feces (g kg^−1^).

### 2.5. Sample Collection

At the start of the feeding trial, 27 turtles (nine turtles as one replicate) were randomly selected and gently dried with a paper towel, and then stored at −20 °C for body phosphorus content analysis to calculate phosphorus deposition rate in whole body. At the end of the trial, turtles were fasted for 24 h, quickly netted and narcotized with 1000 mg/L MS-222 (Green Hengxing Biotech Co., Ltd., Beijing, China) [45]. Subsequently, the turtles from each tank were batch-weighed and counted to assess the growth performance. Afterwards, five turtles from each tank were arbitrarily chosen to analyze whole body composition and bone phosphorus content. Another three turtles per tank were decapitated and blood were collected into heparinized tubes, then centrifuged at 4000× *g*, 4 °C for 15 min, and plasma aliquots were frozen at −80 °C for hematological parameters. After that, the turtles were dissected on chilled trays, the small intestine and liver were collected, frozen in the liquid nitrogen and stored at −80 °C for gene expression analyses. The small intestine was differentiated from the large intestine according to the previous study [46].

### 2.6. Chemical Analysis

The biochemical analyses (moisture, crude protein, crude lipid, crude ash and gross energy) of experimental diets and turtle body were determined in accordance with an earlier study [47]. The total phosphorus and calcium contents of experimental diets, feces, turtle body and bone were determined according to national standards, respectively (GB/T 6437-2018, GB/T 6436-2018). Briefly, the ground samples were pretreated as follows. About 1 g sample was weighed and placed in a kjeldahl flask. A volume of 30 mL of nitric acid was added into the flask and it was boiled carefully on the electrothermal furnace until there were no yellow fumes escaping from the flask. The flask was cooled naturally for a while and 10 mL hydrochloric acid was added, then continuously boiled until the white fumes escaped (do not evaporate to dryness). The solutions were cooled down, 30 mL deionized water was added and the fluid was boiled continuously. The fluid was cooled down again and finally diluted to 100 mL with deionized water in a volumetric flask to detect the phosphorus and calcium content. The total phosphorus concentration was detected by vanadium-molybdenum-yellow-spectrophotometry using a spectrophotometer (Beijing Purkinje General Instrument Co., Ltd., Beijing, China) at 400 nm wavelength (GB/T 6437-2018). Potassium permanganate titration was applied for calcium measurement in the dietary and body calcium (GB/T 6436-2018).

### 2.7. Physiological Assay

The contents of plasma triglyceride, total cholesterol and glucose, as well as alkaline phosphatase activities in plasma and liver were determined by the colorimetric method using commercial diagnostic kits according to the manufacturer’s protocols (Nanjing Jiancheng Bioengineering Institute, Nanjing, Jiangsu, China). The plasma calcium and phosphorus levels were also determined colorimetrically by mercantile kits (Beijing Solarbio Science & Technology Co., Ltd., Beijing, China).

### 2.8. Real Time qPCR Analysis

Total RNA was extracted from small intestine by the Trizol method. The quality control of RNA, reverse transcription of mRNA to cDNA, primers design, PCR amplification procedure, assessment of PCR product and quantitative real time PCR protocol were performed referring to the previous study [48]. The primer sequence, amplicon size, annealing temperature and accession number are presented in Table 2. GAPDH and β-Actin were applied as housekeeping genes to normalize expressions of target genes. The expression levels of these genes were computed based on 2^−ΔΔCt^ method. Six samples were employed for each treatment and each sample was tested in duplicate.

### 2.9. Statistical Analysis

All data are presented as mean values and standard deviation. The parameters of growth, body composition, physiological parameters, phosphorus utilization and intestinal gene expression were analyzed by one way analysis of variance (ANOVA) and Tukey’s multiple comparisons in Statistica 10.0 software after examining the normality of the data with the Shapiro–Wilk test and checking the homogeneity of variance by Levene’s test. The P value with less than 0.05 was considered statistically significant. The regression analysis was conducted in GraphPad Prism 7.0 (GraphPad Software, Inc.), and best model were chosen based on R^2^.

## 3. Results

### 3.1. Growth Performance

The growth performance of turtles fed different phosphorus diets for 60 days is displayed in Table 3. Our findings revealed that there was no dead turtle throughout the feeding period. Turtles in the D1.00 group exhibited significantly higher FBW (*p* < 0.001), WGR (*p* < 0.001), SGR (*p* < 0.001) and PER (*p* = 0.001) compared with other groups. Meanwhile, turtles in the D1.00 treatment displayed the lowest values of FR and FCR among all treatments. However, turtles fed a high phosphorus diet (D1.63) performed worst in terms of all growth and feed utilization parameters. The optimal requirement of dietary available phosphorus level was estimated by the quadratic regression equation as 1.041% (the total phosphorus was 1.80%) based on WGR (Figure 2).

### 3.2. Whole-Body Proximate Composition and Bone Phosphorus Content

The effects of dietary phosphorus levels on the whole-body composition and bone phosphorus content are shown in Table 4. No noticeable difference was discovered in the moisture (*p* = 0.402) and crude protein contents (*p* = 0.845) of the whole body among treatments. The crude lipid and gross energy of the whole body revealed a markedly decreased trend as the dietary phosphorus levels increased, and the turtles fed D1.63 diet had significantly lower values than those fed diets without phosphorus supplementation (*p* = 0.041 and *p* = 0.045). In contrast, the crude ash, calcium and phosphorus of the whole body and bone phosphorus contents were positively related to dietary phosphorus levels, and significantly higher in the D1.63 group than in the D0.88 group (*p* < 0.01).

### 3.3. Apparent Digestibility Coefficients (ADCs) and Phosphorus Utilization

In vivo ADCs of dry matter and phosphorus in the experimental diets and phosphorus utilization parameters are presented in Table 5. The dry matter digestibility was unaffected by dietary experimental diets (*p* > 0.05). The digestibility of phosphorus was negatively correlated with dietary phosphorus levels, and turtles fed D1.63 had a significantly lower coefficient than those fed D0.88 (*p* = 0.017). Interestingly, an upward trend in the phosphorus retention ratio (PRR) was observed as dietary available phosphorus increased from 0.88% to 1.00%, and the ratio then decreased afterwards. Turtles fed D1.00 had the highest PRR, above that of the other groups (*p* < 0.001). More importantly, the turtles fed D1.00 excreted significantly less phosphorus than those fed D1.18 and D1.63 when the same body weight gain was obtained (*p* < 0.05), and it had no statistical difference compared with D0.88 group.

### 3.4. Physiological Parameters

The impacts of dietary phosphorus levels on physiological parameters were assessed in the plasma and liver of turtles (Table 6). The plasma glucose concentration was unaltered by the experimental diets (*p* = 0.254). The plasma triglyceride and total cholesterol contents, as well as plasma alkaline phosphatase (AKP) increased as dietary available phosphorus levels increased from 0.88% to 1.18%, and then fell afterwards. The turtles fed D1.18 had remarkably higher values of the abovementioned indicators than those fed D1.63 (*p* < 0.05). Plasma phosphorus content increased over the dietary range of phosphorus, and which was significantly higher in D1.63 than D0.88 (*p* = 0.009). However, a decreasing tendency was found in plasma Ca^2+^ and hepatic AKP activity as dietary phosphorus increased, and these two parameters were significantly lower in D1.63 than those in D0.88 and D1.00 (*p* < 0.05).

### 3.5. Relative mRNA Expression of Phosphorus and Calcium Transport-Related Genes in Small Intestine

The intestinal transcriptional levels of calcium and phosphorus transport-related genes were affected by different phosphorus diets (Figure 3). In terms of phosphorus transport-related genes, turtles fed D1.63 had significantly lower mRNA expressions in NaPi-IIb and VDR than those fed D0.88 (*p* < 0.05). The relative expression of NaPi-I was unaffected by the experimental diets (*p* > 0.05). Likewise, the calcium transport-related genes (TRPV6, CABL2 and PMCA1) were significantly downregulated in D1.63 compared to their counterparts in D0.88 (*p* < 0.05).

## 4. Discussion

The results in the present study reveal that dietary phosphorus concentration significantly affects growth performance and feed utilization of *P. sinensis*. Inadequate (0.88%) or excessive (1.18% and 1.63%) dietary available phosphorus (AP) caused decelerated growth and inferior feed utilization of turtles. Among the responsive parameters, relative weight gain was more dependable in determining the phosphorus requirement in fish [49]. The AP requirement was determined at 1.041% (total phosphorus (TP) was 1.80%) based on quadratic regression of weight gain rate against varying dietary available phosphorus levels. The result was close to that of some crustaceans, such as *F. indicus* (TP, 1.84%) [36] and *P. trituberculatus* (TP, 1.59–1.68%) [16], but significantly lower than that in the previous study on soft-shelled turtle (3.0%) [40]. This difference maybe be mainly ascribed to the strains, diet formulation and culture conditions in these two studies, which could be glimpsed from the better growth and feed utilization for the 60-day culture period in the present study than those in the previous study for 10-week period.

In this study, the crude lipid content of the whole body exhibited a decreasing tendency with the dietary phosphorus levels and the turtles fed high-phosphorus diet (1.63% AP) showed a lower value than other groups. Prior studies revealed that reduction of body lipid content by a high-phosphorus diet was closely linked to the alteration of lipid metabolism (lipid syntheses and catabolism) rather than feed intake [38,50], which was also verified by the unchanged feeding rate between low-phosphorus group and high-phosphorus group in this study. The plasma lipid metabolites (cholesterol and triglyceride) in the present study were lowest in D1.63 among the treatments, which confirmed more lipid metabolism caused by the high phosphorus diet. The phosphorus content of whole body and bone phosphorus content were positively connected with dietary AP levels in this study. Similar findings were also observed in Indian major carp (*Labeo rohita*) [51], coho salmon (*Oncorhynchus kisutch*) [52], obscure puffer (*Takifugu obscurus*) [53] and largemouth bass (*Micropterus salmoides*) [54]. This indicated that a greater amount of phosphorus entered into the body through intestinal absorption in turtles fed high-phosphorus diet. It was then deposited in the whole body, particularly in the skeleton, which could also be corroborated by increased plasma phosphorus concentration with dietary AP levels in this study. It was interesting to discover that body calcium content was also increased with the increment of dietary AP levels; however, the plasma calcium levels displayed a decreasing tendency. This could be explained by the following: intestinal calcium absorption decreased as the dietary AP levels increased, but more calcium in the blood tended to deposit in the body in the form of hydroxyapatite. However, in previous studies on tilapia [55] and crab [38], blood calcium concentration remained unaltered by high phosphorus diets. This maybe be related with to culture and species: tilapias and crabs could obtain calcium through their gills from the surrounding water to maintain calcium homeostasis in the body, whereas turtles are lung-breathing, and thus mainly dependent on calcium sources in diets.

In this study, apparent phosphorus digestibility in the diet decreased gradually as the dietary phosphorus levels increased. This indicated that when a lesser amount of dietary phosphorus entered the digestive tract, the efficiency of digestion and absorption was higher, which was in line with earlier studies [20,55]. Two pathways for phosphorus absorption exist, an active transcellular phosphorus transport by NaPi-IIb transporters and a second paracellular pathway [56]. When the luminal phosphate concentration is low, active transport is more significant for Pi transport [57]. Previous studies in fish indicated that low-phosphorus diet could upregulate expression NaPi-IIb transporters to increase Pi absorption [58,59]. It could also be confirmed by the progressive decrease in the expression of intestinal NaPi-IIb transporters in the current study.

Feed quality improvement including dietary phosphorus retention efficiency is one of the crucial strategies to develop environmental-friendly aquafeeds [60]. In the present study, the highest phosphorus retention ratio was noticed in turtle fed diet containing 1.00% AP, and that those fed higher and lower dietary phosphorus diets displayed lower phosphorus deposition. This indicates that dietary incorporation of 1.00% AP is optimal for optimizing growth and is environment-friendly. The phosphorus retention ratio (61.53%) was comparatively higher than for some other aquaculture animals, such as *L. vannamei* (20.41–27.66%) [61], *C. auratus gibelio* (22.75–41.41%) [62], *Acanthopagrus schlegelii* (36.50–58.84%) [63], *M. salmoides* (44.43–51.89%) [54]. In addition, the phosphorus loss was also compared among treatments; turtles fed diets containing 0.88% and 1.00% AP had lower phosphorus loss (4.66 and 4.95 g/kg body weight gain) compared with turtles fed high-phosphorus diets (9.15 and 17.92 g/kg body weight gain). This further proved that diets containing 1.00% AP are environmentally friendly. The parameter of phosphorus loss should therefore be considered when assessing feed quality from the perspective of environmental protection.

Tissue-nonspecific alkaline phosphatases are membrane-bound ectoenzymes that hydrolyze pyrophosphate during the process of biomineralization and provide phosphate for the formation of hydroxyapatite [64]. In the present study, the plasma and hepatic alkaline phosphatase (AKP) activities were significantly decreased by the high-phosphorus diet. This was consistent with previous studies [16,38,50,65]. This may be due to the fact that a sufficiently high concentration of phosphate was present in the serum of turtles fed the high-phosphorus diet, satisfying the needs for skeleton mineralization; hence, the AKP activities in blood and liver were lower. The significantly higher phosphorus content in plasma of turtles fed the high-phosphorus diet (D1.63) also corroborates these results.

In rats, a low-phosphorus diet increased calcium absorption, whereas a high-phosphorus diet depressed it [66]. Therefore, we determined the mRNA expression of calcium absorption related key genes. In this study, turtles fed the diet containing 1.63% AP had significantly lower mRNA expression of intestinal TRPV6, CABL2 and PMCA1 than other treatments. These results suggest that a high dietary phosphorus level may inhibit intestinal absorption of calcium in juvenile *P. sinensis*. It is well recognized that vitamin D3 is a key element involved in calcium absorption through binding with its nuclear receptor VDR [45,67], and the relative expression of intestinal VDR markedly decreased in the D1.63 group. The results were similar with a study on laying hens, in which the protein level of duodenal VDR and calbindin were down-regulated by high nonphytate phosphorus in the diet [68]. A notably decreased plasma calcium concentration was also discovered in D1.63 group. Therefore, a high-phosphorus diet might affect skeletal development through reducing intestinal calcium absorption (decreased serum calcium concentration and down-regulated intestinal calcium transport-related gene expression), thus exerting a negative influence on growth performance. Previous studies have demonstrated that high dietary phosphorus has caused bone loss in animals [69,70].

## 5. Conclusions

In conclusion, the present study revealed that dietary phosphorus levels significantly affected growth performance, body composition, phosphorus utilization and calcium absorption in juvenile *P. sinensis*. Based on WGR, the requirement of available phosphorus of juvenile *P. sinensis* was 1.041%; the best growth, highest phosphorus retention rate and less phosphorus loss were obtained at this inclusion level. Meanwhile, it could not be ignored that high-phosphorus diet (1.63% available phosphorus) resulted in an impaired growth rate, feed efficiency, plasma physiological parameters and intestinal Ca and P absorption, as well as higher phosphorus discharge under the 60-day feeding period. From the perspective of practical application, the recommended dose of available phosphorus in diets of juvenile cultured turtles and some endangered turtle species is 1.04% when the diet formulation is design or adjusted. Excessive dietary phosphorus not only resulted in a slower growth rate, but also impaired the health of the turtle.

## Figures and Tables

**Figure 1 animals-12-03101-f001:**
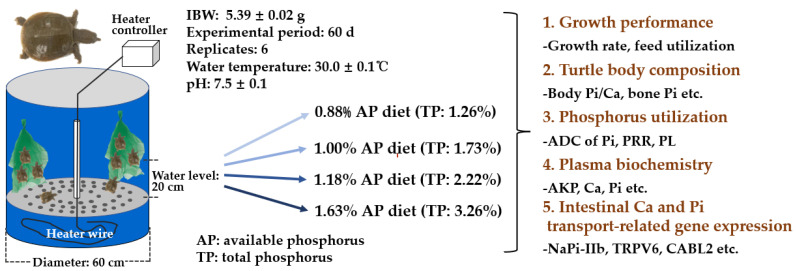
Flow chart of the experimental design.

**Figure 2 animals-12-03101-f002:**
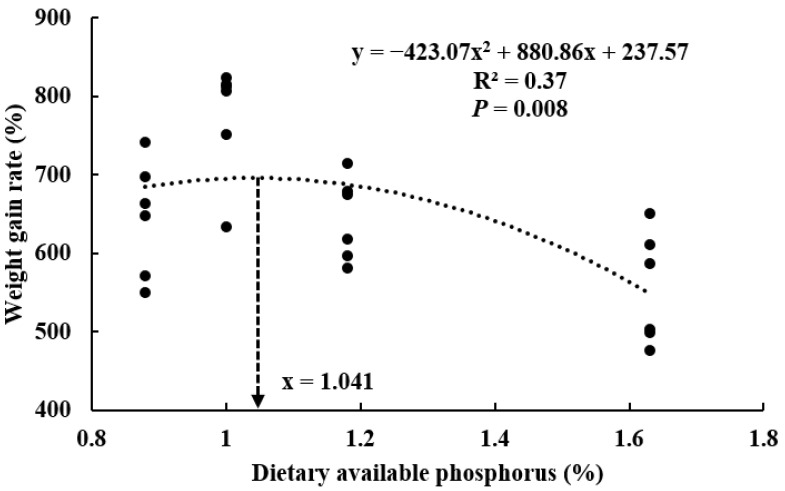
Quadratic regression relationship of weight gain rate against dietary available phosphorus levels.

**Figure 3 animals-12-03101-f003:**
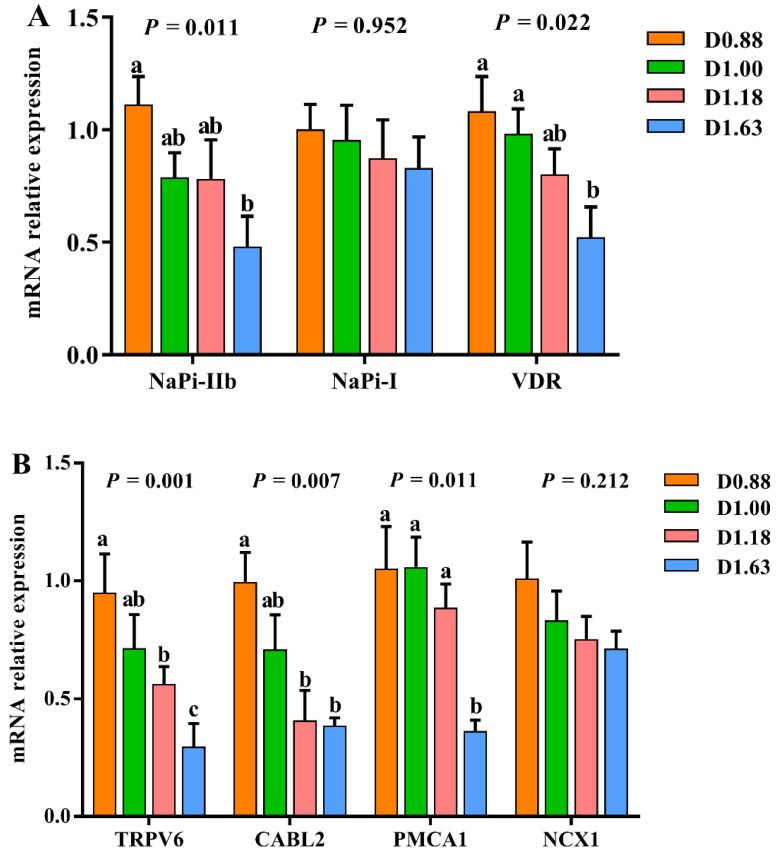
Effects of dietary phosphorus levels on mRNA relative expression of intestinal phosphorus transport-related genes (NaPi-IIb, NaPi-I and VDR) (**A**) and calcium transport-related genes (TRPV6, CABL2, PMCA1 and NCX1) (**B**). Values are listed as mean ± SD (*n* = 6). Different letters indicate significant differences (*p* < 0.05).

**Table 1 animals-12-03101-t001:** Diet formulation and measured macronutrients of experimental diets (%, on DM basis).

Ingredients	MCP0	MCP2	MCP4	MCP8
Chicken meal	18.00	18.00	18.00	18.00
Russian white fishmeal	17.00	17.00	17.00	17.00
Casein	12.00	12.00	12.00	12.00
Wheat gluten	7.00	7.00	7.00	7.00
Squid liver powder	4.00	4.00	4.00	4.00
Yeast powder	4.00	4.00	4.00	4.00
Gelatin	3.00	3.00	3.00	3.00
Yeast extract	2.00	2.00	2.00	2.00
Pregelatinized corn starch	16.00	16.00	16.00	16.00
Fish oil	4.00	4.00	4.00	4.00
Vitamin-mineral premix ^1^	2.00	2.00	2.00	2.00
Others ^2^	1.30	1.30	1.30	1.30
Yttrium oxide	0.10	0.10	0.10	0.10
Microcrystal cellulose	4.22	4.15	3.88	0.53
Limestone	4.38	3.55	2.72	1.07
Monocalcium phosphate	0.00	2.00	4.00	8.00
Proximate composition (%)				
Moisture	16.33	17.69	16.74	16.85
Crude protein	41.14	40.38	40.87	40.84
Crude lipid	4.71	4.88	4.99	5.11
Ash	9.97	10.39	11.65	13.09
Gross energy (MJ/kg)	19.84	19.67	19.47	19.04
Calcium	2.56	2.46	2.46	2.43
Total phosphorus	1.26	1.73	2.22	3.26
Available phosphorus	0.88	1.00	1.18	1.63

^1^ Vitamin-mineral premix: provided by Hebei Haitai Technology Corporation, Hebei, China. ^2^ Others: methionine 0.17, potassium chloride 0.20, sodium chloride 0.10, betaine 0.10, choline chloride 0.10, taurine 0.40, lysine 0.20, mildew inhibitor 0.03.

**Table 2 animals-12-03101-t002:** Primer sequences for quantitative real-time PCR analysis in Chinese soft-shelled turtle.

Genes	Acronym	Primer Sequence (5’-3’)	Amplicon Size (bp)	Annealing Temperature (°C)	Accession No.
Sodium-dependent phosphate transport protein 2b	NaPi-IIb	F-TGTCAAACCCTGTTGCTGGT R-AGGTACCAATGTTTGCCCCC	159	55	XM_025179010.1
Sodium-dependent phosphate transporter 1	NaPi-I	F-GCAGTAAGGCATCCAGTCCC R-ACTGCGGTTCCGAATGAGTT	124	60	XM_006119742.3
Vitamin D receptor	VDR	F-ATGCTCCGCTCCAACCAG R-CGTCCGTGACTTGGTACTTGA	85	58	XM_025179475.1
Transient receptor potential cation channel subfamily V member 6	TRPV6	F-ATCTCCGGGCCATCAAGAAA R-CTAGAGACATTGGCCCCCTT	265	60	XM_006111851.2
Calbindin 2	CABL2	F-GCTGGCTCAAATCCTGCCTA R-CAGGCCTCCATGAATTCGGA	87	60	XM_006113313.3
Plasma membrane Ca^2+^ transporting ATPase	PMCA1	F-GCTTGCTGAATGCACACCAA R-TGACTGCAAGTGGAAGACCC	107	58	XM_006113988.2
Sodium-calcium exchanger 1	NCX1	CATTGCCGCCATCTACCA R-CAGGTCTCCGCCGATAAA	134	58	XM_006112626.3
Glyceraldehyde-3-phosphate dehydrogenase	GAPDH	F-TGGCCCCTCTGGGAAGTTAT R-AGCCATTCCGGTGAGTTTCC	130	55	NM_001286927.1
β-actin	β-actin	F-CTCTTCCAGCCCTCTTTCTT R-TGGCATACAGGTCTTTACGG	106	55	XM_006112915.3

**Table 3 animals-12-03101-t003:** Growth performance of Chinese soft-shelled turtle fed four experimental diets containing different phosphorus levels for 60 days.

Parameters	Experimental Diets				*p* Value
	D0.88	D1.00	D1.18	D1.63	
SR (%) ^1^	100 ± 0.00	100 ± 0.00	100 ± 0.00	100 ± 0.00	-
FBW (g) ^2^	40.18 ± 3.97 ^b^	47.11 ± 3.95 ^a^	40.09 ± 2.85 ^b^	35.27 ± 3.83 ^b^	<0.001
FR (%BW/d) ^3^	2.05 ± 0.07 ^ab^	1.95 ± 0.07 ^b^	2.05 ± 0.06 ^ab^	2.09 ± 0.08 ^a^	0.015
WGR (%) ^4^	645.45 ± 73.67 ^b^	774.03 ± 73.32 ^a^	643.84 ± 52.90 ^b^	554.35 ± 71.00 ^b^	<0.001
SGR (%/d) ^5^	3.34 ± 0.17 ^b^	3.61 ± 0.15 ^a^	3.34 ± 0.12 ^b^	3.12 ± 0.18 ^b^	<0.001
FCR ^6^	0.81 ± 0.05 ^ab^	0.74 ± 0.04 ^b^	0.80 ± 0.04 ^ab^	0.86 ± 0.06 ^a^	0.003
PER ^7^	3.01 ± 0.19 ^b^	3.36 ± 0.18 ^a^	3.04 ± 0.14 ^b^	2.87 ± 0.19 ^b^	0.001

Values are listed as mean ± SD (*n* = 6). Values in the same row with different letters are statistically different (*p* < 0.05). ^1^ SR: Survival rate (%) = (final turtle number/initial turtle number) × 100. ^2^ FBW: Final body weight (g). ^3^ FR: Feeding rate (% BW/d) = 100 × dry feed intake/[days × (FBW × IBW)/2]. ^4^ WGR: Weight gain rate (%) = 100 × (final body weight − initial body weight)/initial body weight. ^5^ SGR: Specific growth rate (%/d) = 100 × [Ln (final body weight) – Ln (initial body weight)]/days. ^6^ FCR: Feed conversion ratio = dry feed intake/fresh body weight gain. ^7^ PER: Protein efficiency ratio = fresh body weight gain/protein intake.

**Table 4 animals-12-03101-t004:** Whole-body proximate composition (wet weight) and bone phosphorus (dry weight) in Chinese soft-shelled turtle fed with different phosphorus diets.

Parameters	Experimental Diets				*p* Value
	D0.88	D1.00	D1.18	D1.63	
Moisture (%)	73.86 ± 1.70	73.68 ± 1.96	72.62 ± 0.20	73.51 ± 0.76	0.402
Crude protein (%)	17.08 ± 0.98	17.00 ± 1.54	17.29 ± 0.28	16.83 ± 0.51	0.845
Crude lipid (%)	4.93 ± 0.69 ^a^	4.53 ± 0.39 ^ab^	4.80 ± 0.53 ^a^	4.06 ± 0.39 ^b^	0.041
Crude ash (%)	3.93 ± 0.38 ^c^	4.61 ± 0.26 ^b^	5.01 ± 0.34 ^ab^	5.31 ± 0.26 ^a^	<0.001
Gross energy (MJ/kg)	5.86 ± 0.38 ^a^	5.68 ± 0.45 ^ab^	5.89 ± 0.37 ^a^	5.26 ± 0.21 ^b^	0.045
Body calcium (%)	0.87 ± 0.03 ^c^	1.03 ± 0.07 ^b^	1.17 ± 0.07 ^ab^	1.26 ± 0.06 ^a^	<0.001
Body phosphorus (mg/g)	5.54 ± 0.32 ^c^	7.17 ± 0.35 ^b^	8.18 ± 0.50 ^a^	8.94 ± 0.35 ^a^	<0.001
Bone phosphorus (%)	7.87 ± 1.26 ^b^	9.00 ± 1.15 ^ab^	10.3 ± 0.15 ^a^	10.47 ± 0.41 ^a^	0.009

Values are listed as mean ± SD (*n* = 6). Values in the same row with different letters are statistically different (*p* < 0.05).

**Table 5 animals-12-03101-t005:** Apparent digestibility coefficients and phosphorus utilization in Chinese soft-shelled turtle fed with different phosphorus diets.

Parameters	Experimental Diets				*p* Value
	D0.88	D1.00	D1.18	D1.63	
ADC_DM_ (%) ^1^	81.42 ± 0.64	83.59 ± 0.70	83.03 ± 1.00	82.23 ± 6.09	0.697
ADC_phosphorus_ (%) ^2^	69.68 ± 4.02 ^a^	57.73 ± 3.33 ^ab^	53.22 ± 4.77 ^ab^	50.13 ± 3.28 ^b^	0.017
PRR (%) ^3^	54.34 ± 3.21 ^b^	61.53 ± 2.96 ^a^	48.96 ± 1.93 ^c^	35.99 ± 2.09 ^d^	<0.001
PL (g/kg weight gain) ^4^	4.66 ± 0.63 ^c^	4.95 ± 0.68 ^c^	9.15 ± 0.76 ^b^	17.92 ± 1.76 ^a^	<0.001

Values are listed as mean ± SD (*n* = 6). Values in the same row with different letters are statistically different (*p* < 0.05). ^1^ ADC_DM_: Apparent digestibility coefficient of dry matter. ^2^ ADC_phosphorus_: Apparent digestibility coefficient of dietary phosphorus. ^3^ PRR: Phosphorus retention ratio (%) = [((final body weight (g) × final body phosphorus content (%)) − (initial body weight (g) × initial body phosphorus content (%))/phosphorus intake (g)] × 100. ^4^ PL: Phosphorus loss (g/kg weight gain) = [phosphorus intake (g) − (final body weight (g) × final body phosphorus content (%) − initial body weight (g) × initial body phosphorus content (%))]/body weight gain (kg).

**Table 6 animals-12-03101-t006:** Plasma physiological and biochemical parameters and liver alkaline phosphatase in Chinese soft-shelled turtle fed with different phosphorus diets.

Parameters	Experimental Diets				*p* Value
	D0.88	D1.00	D1.18	D1.63	
Triglyceride (mmol/L)	3.23 ± 0.58 ^b^	3.58 ± 1.06 ^b^	4.88 ± 0.48 ^a^	3.14 ± 0.38 ^b^	0.007
Total cholesterol (mmol/L)	5.88 ± 0.74 ^ab^	6.09 ± 0.36 ^a^	6.51 ± 0.52 ^a^	4.95 ± 0.81^b^	0.003
Glucose (mmol/L)	5.74 ± 0.49	5.85 ± 0.46	5.43 ± 0.36	5.47 ± 0.57	0.254
Phosphorus (mmol/L)	1.52 ± 0.06 ^b^	1.62 ± 0.10 ^ab^	1.65 ± 0.08 ^ab^	1.73 ± 0.16 ^a^	0.009
Calcium (mmol/L)	1.17 ± 0.17 ^a^	1.19 ± 0.19 ^a^	1.11 ± 0.16 ^ab^	0.87 ± 0.17 ^b^	0.013
Alkaline phosphatase (King units/100 mL)	53.72 ± 12.13 ^ab^	56.22 ± 12.88 ^ab^	63.08 ± 8.52 ^a^	42.07 ± 4.29 ^b^	0.014
Liver Alkaline phosphatase (King units/gprot)	0.45 ± 0.10 ^a^	0.44 ± 0.11 ^a^	0.37 ± 0.09 ^ab^	0.27 ± 0.04 ^b^	0.009

Values are listed as mean ± SD (*n* = 6). Values in the same row with different letters are statistically different (*p* < 0.05).

## Data Availability

Data available on request from the authors.

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
