# Peer review of "Effects of Dietary Phosphorus Levels on Growth Performance, Phosphorus Utilization and Intestinal Calcium and Phosphorus Transport-Related Genes Expression of Juvenile Chinese Soft-Shelled Turtle (Pelodiscus sinensis)"

_animals, 2022, doi:10.3390/ani12223101_

Round 1
Reviewer 1 Report
Dear authors,
Congratulations! Overall, a very interesting. Some minor adaptations can/should be made:
*You forgot the "Simple Summary".
*Your title is affirmative! You need a more nuanced article title.
*Introduction: Your introduction is long and general. You need to contextualize your work in your introduction.
*MATERIALS AND METHODS:
-How did you choose the phosphorus incorporation rates in the different groups?
-Compare the initial body weights of the turtles by tank and by group?
-Did you carry out temperature and bacteriological monitoring in the tanks throughout the experiment?
-Some information about the management of the animals during the experiment is missing.
- photos/graphics explaining the composition of the different experimental groups can help simplify the understanding of the experimental protocol.
-For the statistical analysis, what did you take as experimental unit for the calculation, the tank, the replicate, turtle,...?
-What statistical analysis did you use to link dietary phosphorus effect with growth performance, phosphorus utilization and expression of genes related to intestinal calcium and phosphorus transport?
*Results: check the meaning of your results. For the example, I think that in figure 2, for NCX1, I think we should put letters because there are differences?
*Discussion: It would be interesting to present a cross-sectional discussion where all results are discussed in a linked manner.
Sincerely,
Reviewer 2 Report
The manuscript is not following the journal guidelines in terms of:
1- Absence of Simple Summary.
2- Capitals are not applied for keywords.
3- Citations and references are not as per the journal guidelines.
It is not clear in the introduction the factors that affect the availability of phosphorus.
The conclusion of the introduction section is not satisfactory; please revise it.
Line 123-124: need supporting reference.
Line 126-128: Kindly elaborate on the ingredient processing during feed preparation (Cooking/ Steaming, psi). Is it a sinking pellet of floating/Extruded pellet feed? Since phosphorus is heat labile elaborate on the method followed during feed preparation and storage.
How did you determine the mentioned gross energy in Table 1?
Please refer to the determined quadratic regression relationship in the methodology section.
Reviewer 3 Report
I find your paper well-written, structured, referenced, and supported with graphs. I would add to the methods how turtles were kept (e.g., light sources, UV-light? and for how long the light was on).
You mention in your introduction that turtle consumption has medical value! Please provide this specific statement with scientific references that support your statement. I am aware that "medical values" as justification for exotic animal consumption is most often superstitious and such should not be part of a scientific article.

Round 2
Reviewer 2 Report
The paper is greatly improved and can be accepted for publication.
Author Response
The author have revised the manuscript according to academic editor. Please see the attachment.
